# Comparative Analysis of Visitor Codes of Conduct in Chinese and Anglophone Zoos

**DOI:** 10.3390/ani14243647

**Published:** 2024-12-17

**Authors:** Yulei Guo, David Fennell

**Affiliations:** 1Chengdu Research Base of Giant Panda Breeding, Chengdu 610081, China; 2Geography and Tourism Studies, Brock University, St. Catharines, ON L2S 3A1, Canada; dfennell@brocku.ca

**Keywords:** codes of conduct, visitor management, cross-cultural comparison, visitor engagement

## Abstract

Zoos around the world have rules to guide how visitors should act. While these well-intentioned rules are meant to help keep everything at the zoo running smoothly, the idea of ensuring visitors behave ethically—particularly regarding animal–human interactions and encounters—has not been considered in much detail until now. This research examines 899 statements from 27 zoos in China and 22 zoos in countries where English is spoken, finding that many zoo codes share similar rules and structures. This study shows that existing codes often focus on strict and direct rules to control visitor behavior. However, it suggests that these codes could be improved by placing more emphasis on animal welfare and educating visitors about conservation. By also considering the purpose and specific context of different zoos, this study recommends making these rules more engaging and supportive of better zoo management. This approach could lead to a more informed and involved public, helping zoos achieve their educational and conservation goals more effectively.

## 1. Introduction

Berger [1] (p. 24) writes that the zoo as an institution is a suitable example of the contrast between humans and animals in society. As Kagan, Allard, and Cater [2] (p. 66) suggest, zoos are “one societal reflection of how humans think and feel about other animals”. For many zoos, this reflection translates into an ethical responsibility that shapes guidelines for human behavior and interactions with animals. Among the key elements that convey this responsibility are codes of conduct, which explicitly outline expectations for human behavior toward animals. According to Malloy and Fennell [3] (p. 453), codes of conduct in tourism “represent the culture of an organization and act as a vehicle to communicate the ethical nature of this culture internally to employees and externally to clients and the public at large”. Hence, codes of conduct in tourism establish self-regulatory measures based on moral expectations to ensure visitors’ adherence to these principles [4]. For zoos, codes of conduct, also offered as “visitor guidelines”, “visitor rules”, or a “visitor notice”, provide an initial form of communication between zoos and visitors, which further shapes animal–human interactions. As zoos state, visitor codes of conduct protect people and animals and provide a basic scaffolding for a pleasant experience [5,6]. While codes of ethics are designed to be more value-based in their orientation and thus contain a more philosophical undertone, “codes of conduct or practice are more technical and specific to the actions of an organization or group in time and place” [7] (p. 21). This study employs an exploratory content analysis of visitor codes of conduct published by zoos on official channels to examine how visitor behavior is addressed through ethical appeals. Two foundational ethical perspectives, namely, teleological in the form of “end” and “goal” and deontological for duty and responsibility, are used to explore how the code of conduct zoos published have structured the moral obligations of visitors.

This study on visitor codes in zoos addresses an existing research gap in the current literature. While studies have explored zoos’ mission statements [8,9,10], educational programs [11,12], and websites [13,14,15] in the link between visitors and animals, there is a notable dearth of studies focusing on codes of conduct as a crucial entry point for the moral regulation and management of animal–visitor interactions. As a tool facilitating communication between zoos and visitors, codes employed by zoos also serve as a means of ethical positioning, validating Berger’s (1980) point that zoos are places where contemporary human beings negotiate relationships with animals. Understanding codes of conduct as crucial ethical discourse shaping animal–visitor interactions in zoos, in turn, helps policymakers and management in zoos to draft ethical codes that could influence both the well-being and welfare of human visitors and animals.

In a previous empirical study conducted at Chengdu Research Base of Giant Panda Breeding, it was found that 66.8% of visitors stated that they only skimmed the codes of conduct in less than 3 seconds and tended to violate the codes more frequently than more careful readers [16]. This means that the effectiveness of the codes of conduct is undermined by a lack of meaningful engagement from the majority of visitors, though careful reading of the codes of conduct can yield a positive impact on its readers. This suggests that merely providing the codes, which may be well intentioned and thoughtfully designed, is insufficient to further engage visitors. However, to make this improvement possible, it is necessary to understand the current use and application of codes of conduct in zoos.

This study is the first investigation into visitor zoo codes of conduct from the perspectives of both Chinese and Anglophone nations. The theoretical basis we employ is a combination of two conceptual frameworks. The first includes comparing different codes from deontological and teleological perspectives, while the second further breaks down these codes within a locus of analysis perspectives (local versus cosmopolitan scales). This approach was first used by Malloy and Fennell [7] in their analysis of 40 codes of conduct (414 individual statements) from various Anglophone organizations, such as the Australian Tourist Commission, Antarctica Visitor Guidelines, and the World Wildlife Fund. We adopt this approach to gain further insight into the similarities and differences critical to constructing codes of conduct at animal-based tourism venues in vastly different cultural contexts.

## 2. Literature Review

### 2.1. Codes of Conduct for Visitors

Scholars [17,18] argue that codes of conduct grew in popularity during the late 1980s and can be found in every sector, from nonprofit organizations to large multinationals including professional trade organizations [19]. While most of the research on codes of conduct has been on US-based firms, there has been a trend toward investigating codes in other countries [20]. The purpose of these codes was to initiate measures to mitigate problems related to labor conditions, environmental standards, and human rights but also in response to major scandals in many different types of organizations and sectors, necessitating a more intensive look at ethical behavior.

Fennell and Malloy [7] trace the emergence of codes of conduct in tourism to the United Kingdom’s Country Code in 1951 and show that this early example recognized the rights of local people and the environments upon which they rely. In the 1980s, a surge in tourist codes addressed international tourism development, dealing with local, regional, and national tourist interactions [21]. Fennell and Malloy [7] suggest that codes of conduct in tourism typically target three main groups: industry, tourists, and hosts. As the most significant target audience, tourist codes have been the most well developed [22]. Developing a code of ethics mirrors a tourism organization’s need to regulate behaviors that are of most concern to these organizations. In the case of animals, codes of conduct mediate the interaction between visitors and animals to protect the latter’s needs and interests from the profit and pleasure motives of the tourism industry [23].

The prevalent use of codes of conduct in the tourism industry is a function of the possibility of introducing visitor codes in a relatively short period [24] as well as their effectiveness in curbing depreciative behaviors [22,25,26]. As a management tool, visitor codes are considered a “soft” approach of several different methods [22,27,28] to reduce tourism’s negative impacts. Examples of such usage include disaster tourism [29], war heritage sites [30], wildlife [31,32,33,34,35,36,37], countryside tourism [38], polar tourism [4,39,40], and indigenous tourism [41]. In this latter study, Holmes et al. [41] supported contextualizing the visitor code of conduct by studying indigenous tourism in Lutsel K’e Dene (Denesoline) in Canada. The authors suggest that an indigenized visitor code of conduct is community-based, participatory, and narrative-driven in respecting local voices for autonomy and self-determination [41].

Visitor codes expand on a broad spectrum of technical details because of the industry’s largely fragmented and complex nature. Although designed to affect behavior immediately, codes are also value-laden by applying ethical principles [3,22,42]. According to Fennell and Malloy [7], visitor codes are usually admonitory with a list of “DON’Ts” and “DOs”, and related research by these authors points in the direction of providing more information in codes of ethics rather than too little information [7].

### 2.2. Ethical Imperatives of Zoos and Codes of Conduct

According to Kagan et al. [2], the future of zoos and aquariums begins with the search for a solid moral grounding for enhancing animal–human relationships. This foundation can support zoos in engaging more effectively with their traditional roles, such as education and conservation. Researchers [43] have emphasized that zoos should adapt as dynamic institutions in a changing global society. Spiriti et al. [44] further suggest that incorporating ethical considerations into their social commitment and responsibility can help zoos align their missions with the evolving expectations of conservation, research, education, and entertainment.

Codes of conduct are one of the spaces where zoos communicate their social obligations and responsibilities to the public. Studies have investigated zoos’ mission statements and show that an increasing number of zoos today have prioritized conservation and education [8,45,46,47,48,49], although an equal, if not greater, number of studies have pointed out that entertainment remains the most critical mandate for zoos [49,50,51,52,53]. As Fernandez et al. [51] (p. 1) have noted, zoos’ reputation essentially builds on their ability to entertain and attract visitors before they “struggle to maintain their other goals”. This conflict between the zoo’s focus on entertainment and the need to perform other roles creates the central problem when zoos regulate the behaviors of the public.

In his study of the code of conduct for visitors, Cole [22] suggests that the industry has yet to establish a specific objective of codes aimed at tourists. Cole’s [22] research shows that codes of conduct in tourism are based on enforcing a sense of responsibility rather than providing justification and rationale for the specific behavior. Moreover, Cole [22] shows that the codes’ content and production have received criticism. Scholars have suggested that codes in tourism are mere “green tricks” [54], soft approaches [40,55], futile management facades [56], or excess moral baggage [57]. In terms of the production of codes, researchers are concerned with the lack of public involvement [58] and the risk of sustaining the ethical hegemony of the minority few who were responsible for establishing the codes [57,59]. Cole [22] suggests that the ideal code should initiate conversation and communication between stakeholders, inspire local communities, and disseminate widely among target groups.

Studies investigating the effectiveness of the codes of conduct of wildlife tourism attractions have affirmed the ongoing distrust in codes. Waayers, Newsome, and Lee [60], in a study of turtle tourism in the Exmouth region, Western Australia, reported that 77% of tourists breached the code of conduct. Quiros [61] showed that tourists demonstrated a different level of compliance with the code of conduct in whale shark tourism in the Philippines (e.g., 44% for minimum distance keeping and 99% for no flash photography, no SCUBA, scooters, and jet-skis) and proposed the need for adaptive management focusing on monitoring animal–tourist interactions and flexible, interactive regulations. In their investigation of the Australian National Guidelines for Cetacean Observation, Allen et al. [62] argued that the voluntary code could not manage operators effectively and called for management alternatives. Koroza and Evans [63] pointed out that visitor boats were more compliant with the code of conduct for bottlenose dolphins in West Wales and suggested a greater need for citizen science to enhance the effectiveness of the code.

In recent years, animal associations and institutions have started to publish guidelines for zoo visitor experiences based on previous codes of ethics on animal welfare. For example, the World Association of Zoos and Aquariums (WAZA) [64] recognizes animal–visitor interactions as a significant aspect of animal welfare. It suggests that animals involved in these interactions should receive positive welfare experiences in its Guidelines for Animal–Visitor Interactions. While placing the welfare of the concerned animal at the center, the guideline also considers the safety of both animals and tourists, the evaluation and relevance of the interactive experience, and the dissemination of messages for further conservation actions. WAZA’s guidelines contain a definition of animal–visitor interaction, monitoring and assessment, suitability of animals, staff expertise, messaging, and safety. Zoo and Aquarium Association Australia (ZAAA) [65] proposed in its Animal Visitor Interactions Position Statement that safe, positive, and engaging multispecies experiences prioritize animal welfare and meaningful animal–human connections. Wild Welfare [66], in its position statement on animal–visitor interactions, emphasizes that the interactions should serve as sources of education, meaningful connection, and inspiration while animal welfare is carefully considered.

However, these recent efforts of animal associations and institutions in considering the need to address animal–visitor interactions in zoos and aquariums have yet to be translated into codes of conduct in individual zoos or aquariums. Scholarly studies on codes of conduct in tourism suggest a great risk of producing ineffective codes [16]. Codes need a careful “translation” of the industrial standards into their individual and local contexts to be more effective. As WAZA [64] suggests, the animal–visitor interaction and the effectiveness of the codes of conduct require further evaluation.

### 2.3. Normative and Contextual Ethics in Zoos’ Codes of Conduct

According to Messick and Tenbrunsel [67], codes of conduct in business belong to an area of applied philosophy that bridges the gap between normative and contextual issues. Normative ethics refers to basic principles governing matters such as “how to act, how to live, or what kind of person to be” [68] (p. 2), whereas contextual ethics describe the empirical environment and how the industry operates. Messick and Tenbrunsel [67] suggest that carefully conducted empirical research concerning contextual ethics can provide new insights, data, and principles that enrich normative ethics. Oyeyemi [69] writes that comprehensive codes of conduct must address normative and contextual issues as a reflection on moral groundings, ethical principles, and an understanding of empirical approaches. In response to the recent narrative turn in business studies, Statler and Oliver [70] (p. 91) stress that these codes are “complex, socially embedded sensemaking processes” and propose that codes of conduct can be most effective when they engage deeply with the sensemaking process.

Traditionally, normative ethics has been divided into teleological and deontological ethics (see Table 1). Teleological ethics stresses that the consequences of an action should be valuable and might benefit and please most individuals. Teleological ethics requires constructing a value statement measuring the action’s consequences. The means and intents of the actions are only of secondary importance to the valuable consequences. In contrast, deontological ethics implies a universal law that ensures the consistency between actions and a universal law that governs or guides the action. The deontological approach does not provide a value justification for the consequences of the action but an obligation to follow the rule or perform one’s duty.

The first empirical study to connect normative ethics, codes of ethics, and tourism was conducted by Malloy and Fennell [3] in their study of 40 codes of conduct in ecotourism. These scholars suggest that deontology and teleology are two dichotomous ethical domains needed to understand the plurality of behaviors and their management in tourism. Deontology builds on the premise that duty, responsibility, principles, policies, and procedures are the foundations of visitors’ behavior. With a deontological approach, the means rather than the ends are most central. Priority, therefore, is to ensure customs, traditions, rules, and laws are respected and followed. By contrast, teleology is focused on the ends through behaviors that lead to happiness through the ability to reason (virtue ethics) or seeking the greatest good for the greatest number (utilitarianism). Here, the consequences of one’s actions or inactions and an understanding of the consequences of our actions are of primary importance.

Malloy and Fennell [7] also mapped deontology and teleology on a locus of analysis matrix based on scale. At the local scale, site-based awareness (which includes scales ranging from the local scale right to national) addresses the costs and benefits of tourism to the community. In contrast, the cosmopolitan perspective is configured to incorporate codes of conduct that have application at the global scale, i.e., the code of ethics is not bounded by national or regional perspectives. Malloy and Fennell [3] found that the deontological approach dominates visitor codes even though a teleological approach would be better because it communicates the consequences of performing or not performing acts, i.e., more comprehensive statements were found to be in educating tourists about whether they should abide by a code or not. Interestingly, in the same study, Malloy and Fennell [3] note that statements in codes of conduct concerning animals were more likely to be teleological (52.6%), whereas statements addressing human participants tended to be deontological.

Codes of conduct in zoos also face challenges in normative and contextual issues. Patrick and Tunnicliffe [71], building on the idea of Zoo Voice, suggest that zoo visiting is an experience where Zoo Voice and Visitor Voice merge, with the Zoo Voice describing the zoos’ history and explaining the moral ground of zoos. Codes of conduct are one of the grounds where the Zoo Voice comes together with the Visitor Voice.

Iwuchukwu, Ajang, and Bassey [72] suggest that zoos can justify their ethical position once they align conservation goals with the interest and welfare of individual animals and zoo management. However, they also pointed out that the impact of zoos can be a conflict between the loss of liberty, costs to animal welfare, and impact on the value of animal life as well as the improved welfare and health outcomes of animals, increased attention toward conservation and protection, and the public education and pleasure. As Gray ([73], p. 3) points out, the moral consideration of animals can be influenced by “the complex and intertwined relationship between humans and animals, the nature of animals and their abilities, and a combination of both”. Although researchers suggest that a teleological approach is effective in regulating and guiding the actions of tourists, Gray [73] notes that neither approach has yet to bring satisfying results.

Although zoos have evolved [49], continued managerial vigilance is required to stay relevant to public concerns and sentiments. Given the historical unease surrounding the utilization of animals for human enjoyment and utility [73], examining the contemporary status of ethical codes employed by zoos is imperative.

## 3. Methods

Content analysis was used in this study to investigate the systematic structure and content of zoo codes of conduct. According to Stemler [74], content analysis contains analytical steps such as identifying patterns, themes, or meanings to gain further insight into the data. Camprobí and Coromina [75] point out that content analysis applies objective analytical categories to develop a systematic and consistent analysis of quantified data. This paper aims to show how existing codes of conduct in zoos across Chinese and Anglophone cultures have regulated tourists’ actions and shaped animal–human relationships. The data collected from the field comprise codes of conduct from both Chinese and Anglophone zoos. The key to achieving objectivity in content analysis is the use of a consistent system of categorization.

For this reason, the categorization employed in this study is based on the work conducted by Malloy and Fennell [3], who categorized codes of conduct in tourism based, primarily, on the codes’ philosophical orientation. This paper adapts the category framework of Malloy and Fennell to further explore the animal–human ethics in zoos. The content analysis in this study followed five steps: data collection, coding scheme development, coding, data analysis, and data interpretation.

In the data collection stage, based on investigating differences across cultures in the construction of codes of conduct in zoos, we collected 49 zoo codes of conduct from both Chinese and Anglophone contexts (see Appendix A). For the 27 Chinese zoos, the researchers first investigated their official websites. Some Chinese zoos do not have websites but are accessible through WeChat, a social media account where zoos publish official information. The codes of conduct of Chinese zoos are most often published alongside the ticketing pages as a “Notice to Visitors” (游客须知, youke xuzhi). We conducted extensive searches on Baidu, the mainstream Chinese search engine, and looked for zoos on both local and provincial levels. According to Wang [76], there are more than 200 zoos and wildlife safari parks in China. In the early days of development in China, zoos were found in provincial capitals and direct-administered municipalities [76]. With the keywords “provincial capital” and “zoo”, Baidu’s search engine returned 31 zoos. With the Baidu search, 15 zoos’ visitor codes of conduct were compiled. The rest of the provincial capital zoos did not provide codes of conduct on their official information outlets.

We found that official websites are Anglophone zoos’ most prevalent information sources. Under pages titled “Visitor rules”, “Guest information pack”, “Visitor guidelines”, “Rules”, “Visitor Q&A”, and so on, we collected 22 codes of conduct of zoos from the U.S. (15), U.K. (3), Australia (1), New Zealand (2), and Canada (1). All pages were accessed through Google search using sets of keywords such as “code of conduct”, “nation” and “zoo” (e.g., “visitor rules” + “Australia” + “zoo”). Notably, not all zoos have a visitor rule explicitly published on their official website. For instance, London Zoo’s [77] Q&A section addresses a few questions under the “General Information” category that could be seen as visitor rules. However, the Q&A page of London Zoo was lengthy, potentially containing additional rules dispersed across various sections. Similarly, Victoria Zoo [78] in Melbourne, Australia, has included the codes of conduct on the page “Frequently Asked Questions”. Due to constraints in time and resources, the research team selected textual sections from zoo websites that straightforwardly represented visitor rules. This pragmatic approach aimed to ensure consistency in the type and format of data analyzed. However, it also meant that this study did not undertake an exhaustive search of visitor guidelines across all zoos within each country. Given the priority placed on the accessibility and retrievability of visitor rules on the websites, we refrained from delineating between private and public zoos. Codes from Chinese zoos in Mandarin were compiled into a single document, and codes from English countries were marked as a different case. After deleting irrelevant information (e.g., ticketing policy and locations), the document yielded 344 statements by Chinese zoos and 555 by Anglophone zoos.

During the second stage of coding category adaptation, a bilingual researcher read through and coded all the documents. Despite the Mandarin statements needing to be translated, the codes applied were written in English. We adapted the visitor code of conduct system from Malloy and Fennell [3] into four categories: the orientation of code, the mood of the message, the main focus of the guideline, and philosophical orientation to provide a holistic picture of the use and design of the zoo codes. The focus on zoo tourism allows this paper to skip addressing the “type of tourism” Malloy and Fennell [3] proposed. Also, we recognize that all collected zoo codes were statements developed for visitors and developed by zoos. Malloy and Fennell [3] categorized the main focus of guidelines into six subcategories: people, resource base, other, man-made site, animals, and plants. A preliminary review of the collected data allowed us to drop the category of the man-made site in this study, and the coding process showed that distinctive patterns surfaced within zoo codes of conduct. This observation encouraged us to categorize the codes based on four groups—people, resource base, others, and animals.

The coding process lasted one month, from May to June 2023, on Nvivo 20 for Mac. The initial coding took about 20 days with the help of a research assistant (See Table 2). The 899 statements were coded for their philosophical orientation, the orientation of code, the mood of the message, the purpose of visiting the zoo, and the main focus of the guideline based on the adaptation of the work of Malloy and Fennell [3]:The philosophical orientation explores whether zoos communicated to visitors through a deontological or teleological approach and whether a local or cosmopolitan position had been employed to amplify information on the philosophical approach.The orientation of codes addressed the statement’s content through social, economic, ecological, or combinations of the three aspects.The message’s mood was identified by observing the statement’s use of words and tone. A typical statement in a negative mood employs phrases such as “Do not”, “Keep away”, “Prohibit”, “No”, etc.The main focus of the guideline contains four categories with several subcategories. For animals, we note that zoo codes concern pets and zoo animals. In the category of people, we note codes on human beings’ gatherings, photography, public order, smoking, ticketing, underprivileged, vending, and other activities. Resource base refers to infrastructures and services provided in zoos, signposts, and others. Personal items such as dangerous articles, food and drinks, personal belongings, printed materials, and transportation tools were also important categories in zoo codes. For instance, example 1 in Table 2 was labeled as a local teleological, social and ecological oriented, negative, entertainment-focused, and zoo animal-specific statement. Example 2 was labeled differently for its economic, ecological, and social orientation and primary focus on infrastructure in the resource base.

After labeling all statements, we examined each category and made necessary adjustments. Table 2 provides two examples of how two statements from Anglophone and Chinese zoos were coded. Adjustments and modifications were made to ensure all references were labeled with the correct code. Lastly, a third coder was involved with a 10% random sample of the documents. Table 2 illustrates specific subcategories that emerged after coding the “main focus of guideline” category.

The coded visitor rules were used for further analysis and interpretation during the last stage. The codes enabled a statistical visualization of the visitor rules. The interpretation of the codes was based on comparing and summarizing these results.

## 4. Results

Table 3 summarizes the distribution of five categories we observed from Chinese and Anglophone zoos’ codes of conduct statements. A *t*-test was performed to evaluate whether significant differences between Chinese and Anglophone zoos. The results show that Chinese zoos have deployed moods of message (*p* = 0.000) differently from Anglophone zoos, although, generally, the negative mood dominates both the Chinese (40.1%) and Anglophone (52.6%) messages. Chinese zoos were more likely to adopt a mixed approach (25.6%) than Anglophone zoos (5.2%). A statement in mixed mood often contains a positive suggestion, followed by negative statements.

In contrast, differences between the philosophical orientation (*p* = 0.247), the purpose for visiting the zoo (*p* = 0.227), the orientation of code (*p* = 0.196), and the main focus (*p* = 0.840) are insignificant (Table 2). Following Malloy and Fennell’s [3] discovery, we found that local deontology remains the most widely used philosophical orientation for both Chinese (65.1%) and Anglophone (73.5%) zoos.

The codes of conduct indicate that zoos have predominately regulated visitors’ entertainment over other functions, including animal conservation, scientific research, and education. All zoos have a primary concern over the societal aspects of visitors. We observed similarity (*p* = 0.846) regarding the main focus of the codes in Chinese and Anglophone zoos. Both Chinese (6.4%) and Anglophone (7.5%) zoos have a minor focus on animal–visitor relations.

Figure 1 illustrates the main focus details identified in Chinese and Anglophone zoos. The figure shows that Chinese zoos used codes of conduct to address infrastructure (16.86%) and public order (14.53%). In contrast, Anglophone zoos focused on maintaining public order (22.02%) and the management of zoo animals (10.29%). For zoos in China and the West, people’s actions are the main target of codes. In Figure 1, we note that codes specifically contextualized for zoos included statements concerning animal–human relationships were 10.47% for Chinese zoos and 10.29% for Anglophone zoos.

Figure 2 displays the clustering of philosophical orientations in the four categories. We note that Chinese zoos tended to deploy the local teleological approach when addressing education (100.00%), focusing on economic and ecological (100.00%) and economic (77.78%) orientations, a positive mood (33.90%), and regulating personal items (34.85%). In contrast, Anglophone zoos employed local teleological orientation in education (67.47%), social, economic, and ecological orientation (69.57%), mixed mood (44.83%), and resource base (28.79%). The data show that Chinese and Anglophone zoos tended to justify their educational function rather than take education as an obligation or duty that visitors have to perform. 

Figure 3 illustrates how zoos perceive animal–visitor interactions through codes of conduct. We filtered out “zoo animals” as a subcategory in the main focus and identified eight ways visitors interacted with animals. The figure shows that feeding animals is a mutual and primary concern for both Chinese (21 times) and Anglophone (26 times) zoos. Anglophone zoos promoted respect toward (16) zoo animals. In contrast, Chinese zoos are falling far behind in respect (2), frightening (23), teasing (20), and touching (20) animals, which are also top concerns of all zoos.

## 5. Discussion

The analysis highlights significant similarities in the philosophical orientation, primary focus, orientation of code, and purposes among the codes of conduct for visitors across both Anglophone and Chinese zoos. These codes primarily address the local operation and management of visitors and animals within individual zoo settings. In general, codes of conduct in both contexts emphasize local guidelines, aim to maintain order, and reflect a practical focus on managing human behavior. They often employ a logico-scientific approach to frame ethical visitor behaviors, leading to a shared pattern across zoos. This uniformity underscores a broader consistency in approach rather than emphasizing distinct contextual or cultural variations in the animal–human relationship.

Zoos across cultures prioritize the management of tourist behaviors, with a strong emphasis on maintaining social order within human society. Both Anglophone and Chinese zoos place less emphasis on conveying the significance of the animal–human connection to visitors. Instead, their codes of conduct primarily focus on fostering positive visitors’ social networks, interactions, and social connections. Understandably, zoos are traditionally used as public spaces that serve not only as sites for wildlife conservation and animal breeding but social venues where safe and enjoyable experiences for all visitors are deemed essential. This recognition of zoos as family-friendly and communal spaces where people could socialize, in our study, overshadows the possibility and opportunity for people to connect with animals. Understandably, zoos may rely on other tools such as educational programs and immersive experiences to fully navigate the animal–human connections. However, encouraging and promoting ethical behaviors that respect animals in codes of conduct strategically helps zoos to better align with broader missions of promoting conservation ethics.

Moreover, this study shows that zoo animal–visitor interactions often revolve around activities such as feeding, touching, teasing, and disturbing the animals, with a primary focus on regulating these behaviors. The animal’s presence in the codes of conduct is framed through the lens of managing human actions rather than emphasizing education and conservation [79,80]. While zoos serve as institutions for public engagement and learning, their codes of conduct reflect a predominant focus on human behavior and interactions, aligning with the broader goal of creating structured and enjoyable experiences for visitors.

The absence of significant cultural distinctions between zoos in Chinese and Anglophone societies may have historical roots, as zoos were originally conceived and evolved in modern Anglophone contexts. In China, the establishment and administration of zoos reflected an adoption of this tradition. According to Tang [81], the first Chinese Zoo, “万牲园 Wan Sheng Yuan”, opened in Beijing in 1907 as a response to the nation’s efforts to embrace the Anglophone notion of public and shared urban spaces. As the first public garden in China, Wan Sheng Yuan was envisioned by early modern Chinese elites as a symbol of urban development and a venue for public education.

This historical importance continues to influence the role of zoos in contemporary Chinese society, where their functions and aims align closely with those of Anglophone zoos. This alignment resonates with the findings by Bacon et al. [82], who highlight a shared understanding of animal welfare between Chinese and European zoos. These shared foundations suggest that cultural distinctions have played a limited role in shaping divergent practices, particularly regarding the prioritization of entertainment in zoos worldwide. The broader challenges surrounding animal–visitor interactions in zoos can thus be seen as a part of a universal problem in animal–human relationships.

## 6. Conclusions

In About Looking, Berger [83] observes that animals have become less prominent in human environments over the last two centuries, with zoos serving as structured spaces where their presence is maintained. The study indicates that the role animals play in zoo codes of conduct reflects the evolving responsibilities of zoos in contemporary society. Rees [84] notes that zoos have historically focused on entertainment and visitor engagement, a pattern that continues today. The analysis highlights that animal–human relations described in codes often center on entertainment-focused activities, with limited emphasis on education and conservation-oriented scenarios. Anglophone zoos, in particular, have shown a clear commitment to promoting respect for animals. However, as Fennell [85] suggests, this respect tends to align with deontological principles rather than fostering more interactive or empathetic relationships between humans and animals. For example, zoo codes often focus on advising visitors to refrain from actions such as “feeding, teasing, petting, or touching” rather than suggesting proactive ways to demonstrate respect for animals. We suggest follow-up explanations for the instructions—why feeding, teasing, petting, or touching are not desired by animals—can direct the focus back to animals.

Zoos often adopt a deontological approach in developing their codes, outlining what tourists should or should not do without explicitly addressing the potential outcomes of these actions. Previous research suggests that incorporating a teleological focus—emphasizing the consequences of actions—can enhance visitors’ understanding of their responsibilities at animal-based venues like zoos [3,24]. This perspective could play a role in promoting greater awareness of the broader impact of human behavior on animals, both as individuals and as a species, within anthropogenic contexts [86]. While utilitarianism in animal-based tourism has been critiqued for prioritizing profit and pleasure, it also offers a framework to consider the outcomes experienced by animals. However, the zoo experience often centers on entertainment, which can overshadow other objectives like education and conservation. This dynamic has been described as “constructed care” [85], where institutional priorities shape the care narrative, emphasizing anthropocentric values over an ethic of care grounded in empathy and respect. Zoos balance various institutional realities, often leading to a complex tradeoff between delivering visitor satisfaction and addressing animal welfare [23,87].

The absence of input from other stakeholders (e.g., academics and visitors) may have reinforced the reliance on a deontological approach, reflecting established historical practices. For example, Holmes et al. [88] applied community-based, participatory, and narrative methodologies to create an indigenized visitor code of conduct for Denesoline, Northwest Territories, Canada. The researchers highlighted how involving tourists and local communities in the development process could contribute to more sustainable tourism at the community level. Incorporating diverse stakeholder viewpoints could provide an opportunity for zoos to refine their care philosophies and better align their practices with the broader interests of animals in their facilities. Most importantly, the process could yield more engaging codes of conduct that attract interest and careful reading of visitors that could benefit zoos and their management in turn.

On balance, this study revealed examples of pro-animal messaging. For example, rather than merely instructing visitors not to feed the animals, a few zoos provide explanations about the special and well-balanced diets prepared for zoo animals. The approach shifts the focus from restricting visitors’ feeding actions to fostering an appreciation of the animal’s nutritional needs. By highlighting the benefits of well-balanced diets, these examples create an opportunity for zoos to emphasize animal welfare while engaging visitors in educational experiences. Such initiatives present a mutually beneficial outcome: zoos are actively directing tourists’ behaviors toward animal care, while visitors gain satisfaction from knowing that animals are receiving proper nutrition and that animals benefit from better food. Beyond the utilitarian outcomes of zoo management [89], this study highlights the potential of incorporating teleologically-driven messaging into codes of conduct to enhance educational efforts.

## Figures and Tables

**Figure 1 animals-14-03647-f001:**
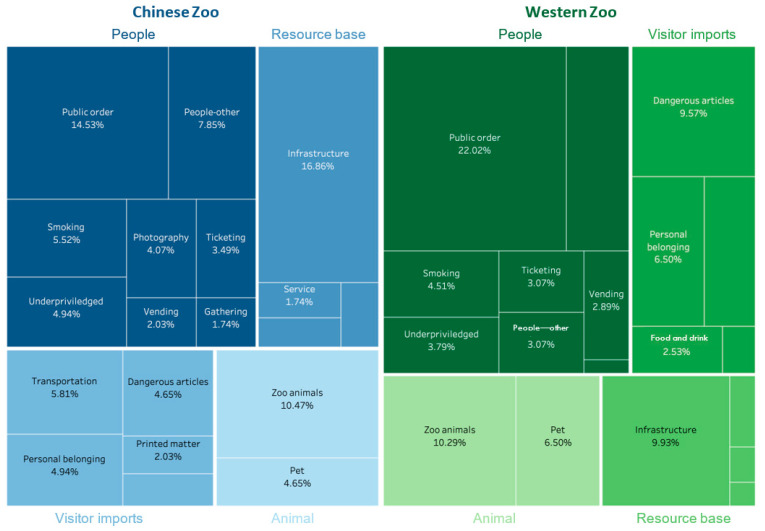
Distribution of the main focus and details in Chinese and Western zoos.

**Figure 2 animals-14-03647-f002:**
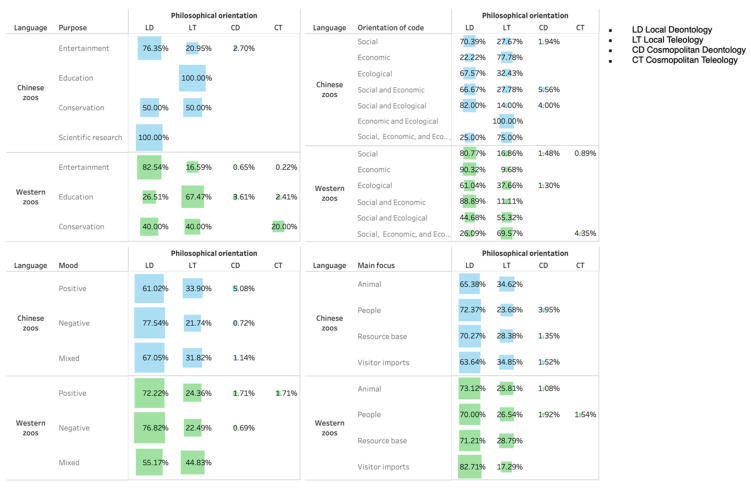
The distribution of philosophical orientations in different categories (by percentage).

**Figure 3 animals-14-03647-f003:**
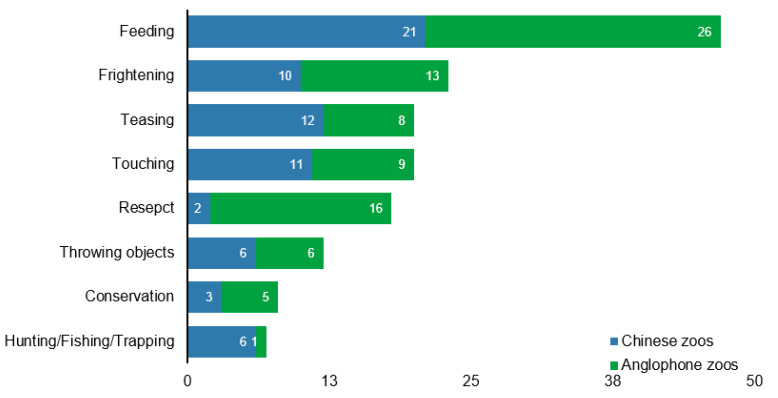
Visitor-zoo animal relations categorized (times) in main focus “zoo animal”.

**Table 1 animals-14-03647-t001:** Typology of theories within normative ethics.

		Premises	Application
Normative ethics	Teleological ethics	Actions judged according to a valuation of their consequences.	The consequences are desired and valued, and most individuals can enjoy some degree of pleasure.
	Deontological ethics	A principle can be good or bad in itself, apart from the values connected with the consequences.	Concepts of right and wrong, rights and duties, and obligations. Customs, traditions, and rules are followed.

**Table 2 animals-14-03647-t002:** Two examples of coded codes of conduct and their categorization (as superscript).

Main Category	Sub Categories			
Philosophical orientation	Cosmopolitan deontology	Cosmopolitan teleology	Local deontology	Local teleology ^1,2^
Orientation of code	Social; Economic; Ecological; Social and ecological ^1^; Social and economic; Economic and ecological; Economic, ecological, and social ^2^			
Mood of message	Negative ^1, 2^	Positive	Mixed	
Purpose of visiting the zoo	Animal conservation	Entertainment^1, 2^	Scientific research	Education
The main focus of the guideline	Animals ^1^	People	Resource base ^2^	Personal items
	Pet Zoo animals ^1^	Gathering Photography Public order Smoking and inflammable Ticketing policy Underprivileged Vending Other	Infrastructure ^2^ Service provision Signs Other	Dangerous articles Food and drinks Personal belongings Printed matter Transportation tools

Example: ^1^. Touching animals is to be avoided unless otherwise noted, including native wildlife within the park. In Hamill Family Play Zoo, some domestic animals are tame, but not all animals can be touched safely. Example ^2^. 园区地形复杂，水系纵横，请勿在水边、陡坡等易险区域玩耍，严禁在禁烟区吸烟或使用明火，携带儿童的游客，请照顾好您的孩子。 (English: The park’s terrain is complex, and the water system is crisscrossed; please do not play near the water’s edge, steep slopes, or other unsafe locations; smoking or using open fire in a non-smoking area is absolutely prohibited; guests with children, please take care of your children. Mandarin: Yuanqu dixing fuza, shuixi zongheng, qingwu zaishuibian, doupo deng yixian quyu wanshua, yanjin zai jinyanqu xiyan huo shiyong minghuo, xiedai ertong de youke, qing zhaoguhao ninde haizi).

**Table 3 animals-14-03647-t003:** Comparison of four categories between Chinese and Anglophone zoos.

	Percent of Total (%)		t-Test	
	Chinese	Anglophone	t	*p*
Philosophical orientation				
Local deontology	69.2	73.5	1.158	0.247
Local teleology	28.5	24.7		
Cosmopolitan deontology	2.3	1.1		
Cosmopolitan teleology	0.0	0.7		
Mood of message				
Negative	40.1	52.6	5.842	0.000
Positive	34.3	42.2		
Mixed	25.6	5.2		
Purpose of visiting the zoo				
Animal conservation	6.4	0.9	1.210	0.227
Entertainment	86.0	84.1		
Scientific research	0.3	0.0		
Education	7.3	15.0		
Orientation of code				
Ecological	4.7	13.9	−1.294	0.196
Social	57.0	61.2		
Economic	2.6	5.6		
Social and economic	10.4	6.5		
Social and ecological	23.2	8.5		
Ecological and economic	0.9	0.0		
Social, ecological, economic	1.2	4.3		
Main focus				
Animal	15.1	16.8	0.194	0.846
People	44.2	47.2		
Resource base	21.5	11.9		
Personal items	19.2	24.1		

## Data Availability

The data presented in this study are available in figshare at https://doi.org/10.6084/m9.figshare.27916560, accessed on 27 November 2024. These data were derived from the websites and official channels of the listed zoos in this study available in the public domain.

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
