# Peer review of "Comparative Analysis of Visitor Codes of Conduct in Chinese and Anglophone Zoos"

_animals, 2024, doi:10.3390/ani14243647_

Round 1

Reviewer 1 Report

Comments and Suggestions for Authors

Whilst the article is well written, it is substantially biased with a strong anti-zoo undertone which makes the research appear to be seeking to undermine zoos rather than to suggest ways of improving the situation.

This could be improved by including a balance of arguments for and against zoos in the introduction and throughout rather than just criticising.

Fundamentally I have issues with whether codes of conduct are the best way of evaluating whether an organisation is meeting conservation and welfare objectives. It seems unsurprising that visitor guidelines and codes of conduct should be human focused and that they don't contain conservation or scientific/research information as this is unlikely to be the purpose of these documents. I would imagine they are more about how visitors should behave during a visit. As such the authors seem very critical of this and use it as an argument against zoos without acknowledging that these documents are taken out of context and that there are likely other documents solely focused on the other aspects (research/welfare/conservation education).

I could see a value in rephrasing the whole article to focus on how codes of conduct in other organisations have an explanatory role on why individuals should behave in a particular way, then demonstrate how zoo codes of conduct currently work before using the discussion to provide recommendations as to how these codes of conduct could be improved to showcase the conservation/research or science work of zoos and explain to visitors the consequences of their actions. 

It would also be interesting to discuss how zoos compare to the other wildlife organisations mentioned and consider are they more human centric or negatively phrased than other organisations.

Other recommendations are as follows:

Ln 45 - there is an extra space in the reference

L 185 - 187 these two sentences seem unrelated - 'tourism having ineffective codes' and 'despite this zoos benefiting from tourism'.

Explanations about teleological and deontological perspectives need to feature much earlier in the introduction and have a stronger thread throughout the whole piece.

Ln 289 - you explain how codes of conduct were sourced. This still needs some comment on how representative the sample is, how and why there is a bias towards US zoos and whether any adjustments were made for the cultural distribution of samples. Why for example is there only one zoo from Australia and one from Canada?

Did the wording as to whether it was a 'code of conduct' or a 'visitor guideline' impact the words used within the document ? - the word 'guideline' indicates a suggested behaviour in contrast to codes of conduct which suggest an enforcible requirement. Was there a noticeable difference?

Where there any other documents that were perhaps ignored such as code of ethics or animal welfare that were also available on the websites or were these code of conducts the only available information? If this information was not examined it needs to be stated that there may have been these other documents available but they were not examined in the research.

Methods - summarise the coding types and examples in a table

The table with main/subcategories is currently unclear - it also has numbers by some phrases e.g., 12 by entertainment and 1 by animals are these references? This is unclear - it would be good to combine this table with a coding table with some more in depth examples of each code.

Ln 381 - you claim that entertainment is more present in these codes rather than education , conservation or research - but I am still struggling to understand why this is a surprise, the codes of conduct refer to visitor behaviour on a day out surely - they would not be expected to contain information about conservation - would they?

If it is normal for codes of conduct to include this more detailed reasoning about conservation, education objectives etc., please explain more in the introduction with examples from how other organisations have done this.

Figure 1 - please add teleological and deontological perspectives to this figure to connect the arguments together.

Figure 2 - why do some aspects have % written and other aspects no %

Discussion - It would be good to structure discussions in combination with recommendations as to how zoos could adapt their current codes of conflict , for example to include more positive phrasing and focus on their other mission areas such as how the actions of visitors impact animal and habitat conservation and scientific research.

Please alter the sentences with substantial anti-zoo bias such as Ln 490 ' holding animals captive - deliberately hidden by research, conservation and education'.

Please be consistent as to whether animal-human or human-animal relationships are used.

Is there any reason that only zoos were considered and not zoos and aquariums?

With a reworking of the argument, this piece could provide useful guidance as to how zoos can improve the relevance of the information presented in codes of conduct. Currently I need more convincing that codes of conduct are an appropriate way of assessing attitudes towards animal welfare and conservation and more examples of how other organisations have effectively used codes of conduct and visitor guidelines in this way.

Author Response

Reviewer 1

Whilst the article is well written, it is substantially biased with a strong anti-zoo undertone which makes the research appear to be seeking to undermine zoos rather than to suggest ways of improving the situation.

This could be improved by including a balance of arguments for and against zoos in the introduction and throughout rather than just criticising.

-Thank you for your thoughtful feedback. We appreciate your perspective on the tone of the article. Our intention is not to undermine zoos, but rather to engage critically with the complex issues surrounding animal welfare in captivity. We have revised the introduction and body of the paper to incorporate a more balanced discussion, highlighting both the positive contributions of zoos and the concerns raised by various stakeholders.

Fundamentally I have issues with whether codes of conduct are the best way of evaluating whether an organisation is meeting conservation and welfare objectives. It seems unsurprising that visitor guidelines and codes of conduct should be human focused and that they don't contain conservation or scientific/research information as this is unlikely to be the purpose of these documents. I would imagine they are more about how visitors should behave during a visit. As such the authors seem very critical of this and use it as an argument against zoos without acknowledging that these documents are taken out of context and that there are likely other documents solely focused on the other aspects (research/welfare/conservation education).

-Thank you very much for this insightful comment. We have made significant modifications throughout the paper to show that we have not used the codes of conduct to measure the whether zoos are meeting conservation and welfare objectives, but to understand how zoos use codes of conduct to regulate visitor behaviors and activities. 

I could see a value in rephrasing the whole article to focus on how codes of conduct in other organisations have an explanatory role on why individuals should behave in a particular way, then demonstrate how zoo codes of conduct currently work before using the discussion to provide recommendations as to how these codes of conduct could be improved to showcase the conservation/research or science work of zoos and explain to visitors the consequences of their actions. 

-Thanks for the comments. This new edition has introduced the use of codes of conduct in other organizations and how they have regulated individuals’ behavior in a particular way. 

It would also be interesting to discuss how zoos compare to the other wildlife organisations mentioned and consider are they more human centric or negatively phrased than other organisations.-Thank you for your valuable feedback. We recognize the importance of comparing zoos with other wildlife organizations or aquariums to provide a more comprehensive perspective. However, our investigation indicates that such comparisons require a thorough understanding of the organizational contexts. For instance, in China, the zoos included in our study are government-established, whereas other wildlife organizations are typically privately owned. This distinction in ownership necessitates a more nuanced approach to comparison, particularly within the Chinese context. Therefore, we chose to focus exclusively on zoos in this paper to ensure a clear and contextually grounded analysis.

Other recommendations are as follows:

Ln 45 - there is an extra space in the reference

-Fixed. 

L 185 - 187 these two sentences seem unrelated - 'tourism having ineffective codes' and 'despite this zoos benefiting from tourism’.

-Revised. 

Explanations about teleological and deontological perspectives need to feature much earlier in the introduction and have a stronger thread throughout the whole piece.

-Yes. The idea of teleological and deontological perspectives have been mentioned earlier in the Introduction and are emphasized throughout the paper. 

Ln 289 - you explain how codes of conduct were sourced. This still needs some comment on how representative the sample is, how and why there is a bias towards US zoos and whether any adjustments were made for the cultural distribution of samples. Why for example is there only one zoo from Australia and one from Canada? 

-Thank you for raising this important point. Due to constraints in time and resources, the research team focused on collecting textual sections from zoo websites that clearly represented visitor rules and were readily accessible. This pragmatic approach aimed to ensure consistency in the type and format of data analyzed. However, it also meant that the study did not undertake an exhaustive search of visitor guidelines across all zoos within each country. Consequently, the sample reflects a selection of zoos that were most accessible and representative of the available online information at the time of data collection, rather than a comprehensive or proportional cultural distribution. While this approach resulted in some geographic imbalances, including a bias toward US zoos and limited representation from countries such as Australia and Canada, it provided a practical basis for exploring cross-cultural patterns in zoo codes of conduct.

Did the wording as to whether it was a 'code of conduct' or a 'visitor guideline' impact the words used within the document ? - the word 'guideline' indicates a suggested behaviour in contrast to codes of conduct which suggest an enforcible requirement. Was there a noticeable difference?

-Thank you for your observation. We found that official websites are Anglophone zoos' most prevalent information sources, while the code of conduct are often listed under pages titled "Visitor rules," "Guest information pack," "Visitor guidelines," "Rules," "Visitor Q&A," and so on. We did not venture to discuss the difference between the code of conduct or the visitor guidelines since this will, obviously, include the discussion of other names as well. 

Where there any other documents that were perhaps ignored such as code of ethics or animal welfare that were also available on the websites or were these code of conducts the only available information? If this information was not examined it needs to be stated that there may have been these other documents available but they were not examined in the research.

-Thank you for this comment. We are certain that there are many animal welfare documents that zoos have published on their website or their official account. However, it was visitor codes of conduct that drew our exclusive attention in this study. For this purpose, we have examined a wide range of documents, as mentioned above (pages titled "Visitor rules," "Guest information pack," "Visitor guidelines," "Rules," "Visitor Q&A," and so on) that specifically addressing the management of visitors. For example, on Woodland Park Zoo, there are webpages and sections devoted to animal conservation and animal welfare such as “Saving wildlife”, “Learn” “Support”, all of which are rich resources in animal welfare and conservation education. The code of conduct, on the zoo’s website, was listed as a part of “Tickets & Hours” section and does not have an individual page at all. This is even more prominent in Chinese zoos, where the codes are often sub-listed under ticketing information. We are very aware of the specific location of the code of conducts along the search and by no mean would suggest that these codes are indicators of animal welfare effort of zoos. 

Methods - summarise the coding types and examples in a table

The table with main/subcategories is currently unclear - it also has numbers by some phrases e.g., 12 by entertainment and 1 by animals are these references? This is unclear - it would be good to combine this table with a coding table with some more in depth examples of each code.

-Thank you. Table 1 provide examples as well as coding types included in this study. The numbers after the phrases indicate their categorization. For example, 12 by entertainment means both example 1 and example 2 concerned entertainment and online example 1 talked about animals. In this version, we have also provided a supplementary table for the main category (not all categories since there are about 40 categories in the full list) including the philosophical orientation and the orientation of code. 

Ln 381 - you claim that entertainment is more present in these codes rather than education , conservation or research - but I am still struggling to understand why this is a surprise, the codes of conduct refer to visitor behaviour on a day out surely - they would not be expected to contain information about conservation - would they?-Thank you for this critical observation. We understand your concern. As we have mentioned, the code of conduct regulate the behavior of visitors. And this also extends to the fact that zoos have, at least indirectly and in this study, perceived zoogoers as visitors seeking for entertainment rather than conservation education and thus, the focus on regulating zoogoers’ entertainment behaviors. We believe it is this papers’ intention that maybe a change is possible when zoos perceive visitors as people seeking education or conservation and hence reorient the code of conduct towards providing guidance on conservation learning.

If it is normal for codes of conduct to include this more detailed reasoning about conservation, education objectives etc., please explain more in the introduction with examples from how other organisations have done this.

-Thank you for your suggestion to include more detailed reasoning about conservation, education, and other objectives in the introduction. While this perspective is valuable, our study’s findings indicate that such detailed reasoning is not commonly reflected in the codes of conduct of the zoos analyzed. Among the approximately 50 zoos included in our study, the primary focus of these codes is on regulating visitor behavior, often emphasizing entertainment-related aspects rather than explicitly addressing conservation or education objectives.

It is also a fact that codes of conduct have historically received less attention from both zoos and academics. This lack of focus has resulted in many codes being relatively similar in content and structure, with limited evolution to incorporate broader institutional objectives like conservation or education.

Moreover, codes of conduct are primarily operational documents intended to guide visitor behavior rather than strategic documents outlining institutional missions. Therefore, examples of codes incorporating conservation or educational reasoning were not available within our dataset. While other organizations may adopt different approaches, such examples fall outside the scope of our study, which focuses specifically on zoos and their visitor-facing materials.

We appreciate your perspective and agree that the broader inclusion of conservation and educational objectives in visitor guidelines could represent an aspirational direction for zoos. However, the absence of such examples in our findings underscores the significance of this study in highlighting current practices and suggesting pathways for improvement.

Figure 1 - please add teleological and deontological perspectives to this figure to connect the arguments together.

-Thank you for this suggestion. We have showed in Table 1 that there are five main categories while teleological and deontological perspectives belong to one of the main categories. Figure 1, however, illustrate another main category, namely, the main focus of the guideline. A cross visualization of the philosophical perspective and the main focus please see Figure 2. 

Figure 2 - why do some aspects have % written and other aspects no %

-Thank you. We have provided a new image that enables better reading and the mark of full percentages.  

Discussion - It would be good to structure discussions in combination with recommendations as to how zoos could adapt their current codes of conflict , for example to include more positive phrasing and focus on their other mission areas such as how the actions of visitors impact animal and habitat conservation and scientific research.

-Thank you, we have made significant modifications in Discussion to provide suggestions and recommendations for zoos to adapt their current codes of conflict. 

Please alter the sentences with substantial anti-zoo bias such as Ln 490 ' holding animals captive - deliberately hidden by research, conservation and education’.

-OK!

Please be consistent as to whether animal-human or human-animal relationships are used.

-OK. Now we are using “animal-human” across the paper. 

Is there any reason that only zoos were considered and not zoos and aquariums?

-Thank you for this critical observation. We have not included aquariums in this study, as explained above, due to the different ownership these venues often claim in the Chinese contexts. We are aware that private company might have a slightly different approach to animal welfare and thus the main focus on the zoos. 

With a reworking of the argument, this piece could provide useful guidance as to how zoos can improve the relevance of the information presented in codes of conduct. Currently I need more convincing that codes of conduct are an appropriate way of assessing attitudes towards animal welfare and conservation and more examples of how other organisations have effectively used codes of conduct and visitor guidelines in this way.

-Thank you. We have made more significant revisions in the conclusion and tried to provide more examples of how other organizations have used code and guidelines more effectively. 

Reviewer 2 Report

Comments and Suggestions for Authors

1.General reception of the manuscript

The article entitled "Comparative analysis of visitor codes of conduct in Chinese and Anglophone zoos" is a detailed analysis of visitor behavior codes in zoos in China and English-speaking countries.

The aim of the study is to identify differences and similarities in the approach to the management of ethical behaviour in the context of zoo visitors. The innovation of the study lies in the juxtaposition of deontological and teleological approaches to code management and the analysis of codes at the cultural and regional levels, which enriches the study of ethics in tourism.

The results indicate a marginalization of animal welfare issues in the codes and a strong focus on order regulations and entertainment aspects, which, as the authors state, is particularly visible in the case of English-language gardens.

2. Assessment of the consistency and correctness of the:

A) Subject: The title of the article reflects its content well. The phrase "Comparative analysis of visitor codes of conduct" clearly suggests the scope of the analysis. It also describes well the discussed aspects of the study, such as cultural differences and the ethical management of visitor behavior in a comparative context [1–3].

B) Purpose and hypotheses: The article clearly distinguishes the purpose of the study and formulates hypotheses about the differences in approaches to ethics and moral values represented in the codes of Chinese and English-language gardens. The aim was outlined in the first paragraphs of the introduction, and directly refers to the topic of the work [33–42].

C) Research methods: The method of content analysis is properly selected for the purpose of the study. The authors used quantitative and qualitative analyses, including statistical tests (t-test), which allow to grasp differences in tone, philosophical orientation and the main goals of the codes. The description of the method is detailed and includes an explanation of all stages of the analysis – from data collection to coding and interpretation [260–326]. However, it would be necessary to indicate more precisely the reasons for the selection of individual variables, which would give a full picture of the analysis.

D) Research material: The selection of research material has been carried out carefully, taking into account the availability of codes in both Chinese and English. Diverse sources of information for both groups of gardens are also described – websites and social media in the case of Chinese gardens and official websites for English-language gardens [277–308]. The number of codices (49) is adequate for a comparative analysis, but the material could be expanded to include additional cases from other cultures to increase the generality of the results, such material is sufficient for this analysis, but too modest for such far-reaching and unambiguous conclusions.

E) Results: The results were presented in a clear way, with appropriate support of tables and charts. The results indicate a clear predominance of deontological orientation in the codes of both groups of gardens, however, the English-language codes show a stronger focus on regulating visitor behavior than Chinese codes, where a mixed approach is more often used [364–403]. This is consistent with the adopted hypothesis of the study and is reflected in the conclusions.

F) Charts, tables, figures: Charts and tables have been prepared in accordance with standards and illustrate the results well. Tables 2 and 3 clearly present the distribution of philosophical orientations and the goals of the codices in both groups of gardens. However, there are no detailed captions that would facilitate their interpretation without the need to refer to the content of the text [374–391].Figure 2 The graphic reception is bad and the content is illegible, I recommend it to make it more attractive for better reception and readability of information.

G) Conclusions: Conclusions are logically related to the results and accurately summarize the data obtained. The paper addresses the initial hypotheses well and suggests that the marginalization of animal welfare issues in the codes of both groups of gardens is a consequence of the anthropocentric structure of the organization of these codes [414–471]. However, in my opinion, they are too far-reaching considering the amount of data and the collected research material.

3. Evaluation of the introduction and discussion

The introduction introduces the reader to the subject of the study in a clear way, emphasizing the role of zoos as places where various goals are pursued – from entertainment to education and conservation. References to the literature are reliable and well related to the topic of the study. The introduction and discussion consistently address the issue of animal marginalization and indicate the need to redefine ethical codes to better respond to contemporary challenges related to animal welfare [20–32], [484–516].

The discussion effectively refers to the results obtained in the study, but at times there is a lack of a smooth transition between discussing the results and interpreting them in the context of the literature. In verse [415], the authors emphasize the similarities in the Chinese and English-language codices, which is a good starting point, but the development of this point could be more fully accomplished by a more comprehensive interpretation of these similarities. It would therefore be worthwhile to add examples of specific elements of the codes that confirm the similarities in question, rather than merely emphasizing their presence.

The authors note a strong anthropocentric approach in the analyzed codices [429–431], which is an accurate observation, but it has not been sufficiently developed. I encourage you to develop this aspect by analyzing the reason for the dominance of the anthropocentric approach in codices, both in the context of Chinese and English-speaking culture. There is also no discussion about how the codes could be modified to include a pro-animal perspective.

The discussion indicates that the dominance of the deontological approach [478] may limit the effectiveness of codes in promoting positive human-animal relationships. While this observation is important, the authors do not analyze in detail what elements of the teleological approach could increase the effectiveness of the codes. Alternative proposals – such as more practical guidance or an emphasis on the consequences for animals – could provide valuable added value to this part of the chapter.

The authors raise the issue of similarities between Chinese and English-language codices, arguing that they result from China's adoption of the English-language model of the zoo [447–455]. At this point, we could develop considerations on how cultural differences could affect the content of codices and why, despite these differences, codices remain similar to each other. There is also no reference to how cultural differences could enrich the codes by introducing specific rules tailored to local values and social norms.

The discussion only superficially refers to the marginalization of animals in the codes [462–465]. The authors rightly note that animals in codices are often depicted as objects to be seen, with minimal emphasis on their welfare. However, it would be worthwhile to elaborate on this topic, especially in terms of the effects of such an approach on the perception of animals by visitors. recommends adding analyses or suggestions on how to increase the visibility of animals as subjects with their own needs and values in the content of the codes.

The discussion emphasizes the importance of the results and refers to the literature, pointing to the need to include more stakeholders in the development of codes. The examples cited are well chosen and provide solid support for arguments in favor of a more diverse ethical approach [501–505].

 4. Typing, spelling, language errors and incorrect entries in the Literature chapter

·       [61]: A typo in the word "interaction".

·       [235]: No comma after the word "acts".

·       [139]: You should add "to" before "justify" for grammatical correctness.

·       [47]: Repetition of "codes of conduct or practice" – the notation can be simplified.

·       [527]: The abbreviated form "Agyeman & Asebah" should be extended to full names if the full form for other authors is used.

·       [692]: No comma after „Roe et al.”

·       [617]: A period at the end of a sentence is missing from the reference "Leopold, T. (2007)".

6.Summary: The manuscript is consistent and logical, and the results of the study enrich the knowledge about the ethical management of the behavior of zoo visitors. Identifying the marginalisation of animals in the codes and proposing a more differentiated approach could have a significant impact on further research and practice in this area.

After removing the errors and adapting the manuscript to the suggestions contained in the text, I recommend the article for publication, as interesting to start an academic discussion on the issues presented.

Author Response

Reviewer 2

1.General reception of the manuscript

The article entitled "Comparative analysis of visitor codes of conduct in Chinese and Anglophone zoos" is a detailed analysis of visitor behavior codes in zoos in China and English-speaking countries.

The aim of the study is to identify differences and similarities in the approach to the management of ethical behaviour in the context of zoo visitors. The innovation of the study lies in the juxtaposition of deontological and teleological approaches to code management and the analysis of codes at the cultural and regional levels, which enriches the study of ethics in tourism.

The results indicate a marginalization of animal welfare issues in the codes and a strong focus on order regulations and entertainment aspects, which, as the authors state, is particularly visible in the case of English-language gardens.

-Thank you for your detailed summary. 

2. Assessment of the consistency and correctness of the:

  1. Subject: The title of the article reflects its content well. The phrase "Comparative analysis of visitor codes of conduct" clearly suggests the scope of the analysis. It also describes well the discussed aspects of the study, such as cultural differences and the ethical management of visitor behavior in a comparative context [1–3].

-Thank you. 

B) Purpose and hypotheses: The article clearly distinguishes the purpose of the study and formulates hypotheses about the differences in approaches to ethics and moral values represented in the codes of Chinese and English-language gardens. The aim was outlined in the first paragraphs of the introduction, and directly refers to the topic of the work [33–42].

-Thank you. 

C) Research methods: The method of content analysis is properly selected for the purpose of the study. The authors used quantitative and qualitative analyses, including statistical tests (t-test), which allow to grasp differences in tone, philosophical orientation and the main goals of the codes. The description of the method is detailed and includes an explanation of all stages of the analysis – from data collection to coding and interpretation [260–326]. However, it would be necessary to indicate more precisely the reasons for the selection of individual variables, which would give a full picture of the analysis.

-Thank you. We have returned to the Method section to refine the structures and details of for the selection of the individual variables to give a full picture of the analysis. 

D) Research material: The selection of research material has been carried out carefully, taking into account the availability of codes in both Chinese and English. Diverse sources of information for both groups of gardens are also described – websites and social media in the case of Chinese gardens and official websites for English-language gardens [277–308]. The number of codices (49) is adequate for a comparative analysis, but the material could be expanded to include additional cases from other cultures to increase the generality of the results, such material is sufficient for this analysis, but too modest for such far-reaching and unambiguous conclusions.

-Thank you for your thoughtful feedback regarding the research material. Expanding the study to include additional cases from other cultures is indeed an excellent suggestion and something we would love to pursue in future research. We have tried in a few cases. For example, the search of other European zoos showed that language barrier was the first problem we need to handle. Also, at this stage, the scope was limited by available resources and the number researchers involved. While the selected material provides a sufficient foundation for the comparative analysis, we acknowledge that broader inclusion could enhance the generality of the results and allow for more comprehensive conclusions. We hope to address this limitation in subsequent studies as resources and opportunities allow and would love to expand this work into a global investigation. Also, we believe machine learning and text mining can be useful techniques applied to conduct future research, which is a task in our future research plan. Thank you again for your valuable input!

E) Results: The results were presented in a clear way, with appropriate support of tables and charts. The results indicate a clear predominance of deontological orientation in the codes of both groups of gardens, however, the English-language codes show a stronger focus on regulating visitor behavior than Chinese codes, where a mixed approach is more often used [364–403]. This is consistent with the adopted hypothesis of the study and is reflected in the conclusions.

-Thanks. 

F) Charts, tables, figures: Charts and tables have been prepared in accordance with standards and illustrate the results well. Tables 2 and 3 clearly present the distribution of philosophical orientations and the goals of the codices in both groups of gardens. However, there are no detailed captions that would facilitate their interpretation without the need to refer to the content of the text [374–391].Figure 2 The graphic reception is bad and the content is illegible, I recommend it to make it more attractive for better reception and readability of information.

-Thank you. For the interpretation of Table 2 and Table 3, we have made modifications to the interpretations in the text in order to provide better access to the texts. 

We have put in a new Figure 2 to afford better reception and readability of information (the change of colors and full percentage marks). 

G) Conclusions: Conclusions are logically related to the results and accurately summarize the data obtained. The paper addresses the initial hypotheses well and suggests that the marginalization of animal welfare issues in the codes of both groups of gardens is a consequence of the anthropocentric structure of the organization of these codes [414–471]. However, in my opinion, they are too far-reaching considering the amount of data and the collected research material.

-Thank you for your valuable feedback. We have taken specific care in revising the paper to ensure that the claims are well-supported by the data and analyses presented. Please see our significantly revised Introduction and Conclusion. We acknowledge that zoos in other cultures and languages are conducting excellent work that warrants further study and could provide new perspectives that might refine or even challenge our conclusions. However, given the scope of the current research and the availability of resources and material, we believe this study provides a meaningful starting point for examining the anthropocentric structure of animal welfare codes. We hope that future research can build on this foundation to include a broader range of cases, offering deeper insights into this important issue.

3. Evaluation of the introduction and discussion

The introduction introduces the reader to the subject of the study in a clear way, emphasizing the role of zoos as places where various goals are pursued – from entertainment to education and conservation. References to the literature are reliable and well related to the topic of the study. The introduction and discussion consistently address the issue of animal marginalization and indicate the need to redefine ethical codes to better respond to contemporary challenges related to animal welfare [20–32], [484–516].

-Thanks. 

The discussion effectively refers to the results obtained in the study, but at times there is a lack of a smooth transition between discussing the results and interpreting them in the context of the literature. In verse [415], the authors emphasize the similarities in the Chinese and English-language codices, which is a good starting point, but the development of this point could be more fully accomplished by a more comprehensive interpretation of these similarities. It would therefore be worthwhile to add examples of specific elements of the codes that confirm the similarities in question, rather than merely emphasizing their presence.

-Thanks, we have added the needed details. 

The authors note a strong anthropocentric approach in the analyzed codices [429–431], which is an accurate observation, but it has not been sufficiently developed. I encourage you to develop this aspect by analyzing the reason for the dominance of the anthropocentric approach in codices, both in the context of Chinese and English-speaking culture. There is also no discussion about how the codes could be modified to include a pro-animal perspective.

-Thanks, we have further developed our observation and explained the current focus of the anthropocentric approach. Possible improvements have been offered. 

The discussion indicates that the dominance of the deontological approach [478] may limit the effectiveness of codes in promoting positive human-animal relationships. While this observation is important, the authors do not analyze in detail what elements of the teleological approach could increase the effectiveness of the codes. Alternative proposals – such as more practical guidance or an emphasis on the consequences for animals – could provide valuable added value to this part of the chapter.

-Thank you! This is a great insight and gives direction for many future research. We think to really understand what elements of the teleological approach could increase the effectiveness of the codes demands further experimental studies or more surveys. Within the limited range of this study, we do not think a collection of codes and their texts could really help us answer these great questions. We would love to, however, address these interesting questions in future studies, which are built on this analysis. 

The authors raise the issue of similarities between Chinese and English-language codices, arguing that they result from China's adoption of the English-language model of the zoo [447–455]. At this point, we could develop considerations on how cultural differences could affect the content of codices and why, despite these differences, codices remain similar to each other. There is also no reference to how cultural differences could enrich the codes by introducing specific rules tailored to local values and social norms.

-Thank you again for raising this thought-provoking point. While we appreciate the importance of examining how cultural differences might influence or enrich the content of zoo codes of conduct, this specific question extends beyond the scope of the current study. Our focus here is on identifying and analyzing the similarities between Chinese and English-language codices and interpreting these within the context of global zoo management practices.

-The exploration of how cultural differences could shape or diversify codices is indeed a valuable direction for future research. A dedicated study could delve into the interplay between cultural values, social norms, and code design, offering deeper insights into how localized approaches might contribute to a more nuanced and context-sensitive visitor management strategy. We acknowledge the significance of this perspective and hope to address it more comprehensively in subsequent work.

The discussion only superficially refers to the marginalization of animals in the codes [462–465]. The authors rightly note that animals in codices are often depicted as objects to be seen, with minimal emphasis on their welfare. However, it would be worthwhile to elaborate on this topic, especially in terms of the effects of such an approach on the perception of animals by visitors. recommends adding analyses or suggestions on how to increase the visibility of animals as subjects with their own needs and values in the content of the codes.

-Thanks. We have added more details and how to increase the visibility of animals in this section. 

The discussion emphasizes the importance of the results and refers to the literature, pointing to the need to include more stakeholders in the development of codes. The examples cited are well chosen and provide solid support for arguments in favor of a more diverse ethical approach [501–505].

Thanks. 

 4. Typing, spelling, language errors and incorrect entries in the Literature chapter

·       [61]: A typo in the word “interaction".

-corrected

·       [235]: No comma after the word “acts".

-corrected

·       [139]: You should add "to" before "justify" for grammatical correctness.

-corrected

·       [47]: Repetition of "codes of conduct or practice" – the notation can be simplified.

·       [527]: The abbreviated form "Agyeman & Asebah" should be extended to full names if the full form for other authors is used.

-corrected

·       [692]: No comma after „Roe et al.”

·       [617]: A period at the end of a sentence is missing from the reference "Leopold, T. (2007)".

6.Summary: The manuscript is consistent and logical, and the results of the study enrich the knowledge about the ethical management of the behavior of zoo visitors. Identifying the marginalisation of animals in the codes and proposing a more differentiated approach could have a significant impact on further research and practice in this area.

-Thanks a lot. 

After removing the errors and adapting the manuscript to the suggestions contained in the text, I recommend the article for publication, as interesting to start an academic discussion on the issues presented.

Reviewer 3 Report

Comments and Suggestions for Authors

Dear authors,

Thank you for such a thought provoking and interesting approach. I really enjoyed reading through the manuscript and have definitely opened my perspective to this new (to me) area of codes of conduct.

Although I don't have much to say about the methodological choices and results, I do have to point out that it seems to me that the authors present a very marked initial framing, denoting an inclination towards some implicit criticism. With this, I would point out that I don't think the first paragraph makes much sense for the study, without the authors wanting to tell a particular story.

Another point I'd like to make is that, despite all the framework and relevance (still little explored in zoos) of the codes of conduct, it seems clear that the authors expect these to go beyond their original function: to model the visitor's behavior during the visit. The authors point out in the discussion that these codes should be more focused on what visitors can do more for the animals and for conservation. With this, I think the authors get into tricky territory, as this is not, for me, the function or purpose of these rules. In order for them to be able to claim this, we must first understand whether the codes are read and by what percentage of visitors, so that it can be considered a tool that contributes to that objective. Next, and as is well described in the manuscript, many of the codes are not highlighted on the respective platforms (not to mention webpages, for obvious reasons), which indicates that the zoos' strategy is not to disseminate them but, and this is my interpretation, to protect themselves legally.

With this in mind, the penultimate paragraph of the discussion also seems a bit forced to me. As it was possible, as mentioned, to construct a code as mentioned in the study by Holmes et al. (2019), it should be noted that this code refers to an indigenous reserve, where culture and respect for local identity prevail, and by this I mean that it is a space inhabited by humans. Of course, it's always easier to work collaboratively with indigenous populations towards a common goal. I believe that this transposition to zoos becomes more difficult since, as the authors mention, the conservation objectives and interests of the animals in captivity are not taken into account. From my knowledge, animals are really not good at speaking our language. On top of this, as far as I know, zoos are the best places for the presence of experts who know the animals' best interests and wellbeing. Just an extra reflection...

Other minor comments:

Either I missed something or the authors mention 27 Chinese zoos (in the text and in the appendix) but then mention that only 15 were analyzed. I think clarification is needed.

In the results (line 378), the authors mention that differences are “more challenging to observe”. They are not challenging... there are simply no differences, as the analysis indicates.

Author Response

Reviewer 3

Dear authors,

Thank you for such a thought provoking and interesting approach. I really enjoyed reading through the manuscript and have definitely opened my perspective to this new (to me) area of codes of conduct.

-Thank you for your kind words. 

Although I don't have much to say about the methodological choices and results, I do have to point out that it seems to me that the authors present a very marked initial framing, denoting an inclination towards some implicit criticism. With this, I would point out that I don't think the first paragraph makes much sense for the study, without the authors wanting to tell a particular story.

-Thank your for this note. In this revision, we have specifically taken care of the initial framing and tried to be constructive in the framing. The first paragraph has been removed in this edition. 

Another point I'd like to make is that, despite all the framework and relevance (still little explored in zoos) of the codes of conduct, it seems clear that the authors expect these to go beyond their original function: to model the visitor's behavior during the visit. The authors point out in the discussion that these codes should be more focused on what visitors can do more for the animals and for conservation. With this, I think the authors get into tricky territory, as this is not, for me, the function or purpose of these rules. In order for them to be able to claim this, we must first understand whether the codes are read and by what percentage of visitors, so that it can be considered a tool that contributes to that objective. Next, and as is well described in the manuscript, many of the codes are not highlighted on the respective platforms (not to mention webpages, for obvious reasons), which indicates that the zoos' strategy is not to disseminate them but, and this is my interpretation, to protect themselves legally.

-We appreciate your insight regarding the original purpose of codes of conduct in zoos and the challenges in extending their scope beyond guiding visitor behavior. While we agree that understanding whether these codes are read and utilized by visitors is a crucial step, our intention was to highlight the untapped potential of these codes as tools for promoting conservation and animal welfare awareness.

We acknowledge the possibility that many zoos may prioritize these codes as legal safeguards rather than educational or behavioral instruments. This observation aligns with our findings that many codes are not prominently displayed on digital platforms. However, we see this as an opportunity to encourage zoos to reconsider how these codes might serve dual purposes—ensuring legal protection while also engaging visitors more actively in conservation efforts.

We have clarified these points in the manuscript to ensure that our argument does not overstep the evidence currently available while proposing future research directions that address the issues you have outlined, such as assessing the visibility and effectiveness of these codes in influencing visitor behavior. We had simultaneously conducted an empirical study showing that only one third of visitors were willing to read the codes of conduct in details (another work already published). These careful readers reported fewer misconducts as well. It means that if the codes are written more engagingly and empathetically, visitors, animals, and zoos could benefit significantly from these codes. This is main purpose of this paper.   

With this in mind, the penultimate paragraph of the discussion also seems a bit forced to me. As it was possible, as mentioned, to construct a code as mentioned in the study by Holmes et al. (2019), it should be noted that this code refers to an indigenous reserve, where culture and respect for local identity prevail, and by this I mean that it is a space inhabited by humans. Of course, it's always easier to work collaboratively with indigenous populations towards a common goal. I believe that this transposition to zoos becomes more difficult since, as the authors mention, the conservation objectives and interests of the animals in captivity are not taken into account. From my knowledge, animals are really not good at speaking our language. On top of this, as far as I know, zoos are the best places for the presence of experts who know the animals' best interests and wellbeing. Just an extra reflection…

-Thank you. We agree that zoos are already the best places for the presence of experts of animals’ best interests and wellbeing. We think the engagement of visitors in designing and writing of the codes can be crucial to prompt the readership of these codes. :) We agree that animals are not too good at speaking our language and might fail to do so, but the inclusion of visitors can perhaps make a more significant difference in terms of managing them. Hence, we use the study of Holmes et al. (2019) to suggest the inclusion of visitors when designing and writing the visitor codes. 

Other minor comments:

Either I missed something or the authors mention 27 Chinese zoos (in the text and in the appendix) but then mention that only 15 were analyzed. I think clarification is needed.

-Thanks for the careful read! We have returned to the text and made necessary modification. 27 Chinese zoos were included, 15 of which were tracked through Baidu search engine. 

In the results (line 378), the authors mention that differences are “more challenging to observe”. They are not challenging... there are simply no differences, as the analysis indicates.

-Thank you. We have made the revision. 

Round 2

Reviewer 1 Report

Comments and Suggestions for Authors

I would like to thank the authors for their significant restructuring and rephrasing of the article. The latest version is a significant improvement and I am very happy that the recommendations have been successfully addressed.

I note only the following (predominantly Typos) corrections:

Ln144: Place not pace

Ln 485: published along_ should be published alongside

Ln 486: define Baidu e.g., a search engine

Ln 497: space needed after U.S. and U.K.

Ln 499-506: please reference the websites and access dates of London Zoo and Zoos Victoria

Table 1: this is still not presenting clearly - I will mention it to editors as it may be something in the processing and outside of your direct control

Figure 1: much clearer

Figure 2: could you explain LD, LT, CD and CT in the legend.

Well done on all the improvements. Once the above minor edits have been addressed I see no further issue with the paper.

Author Response

Dear reviewer, 

Thank you very much for your kind words. We have corrected the points you have listed. 

We formatted Table 1 as well to make it more readable, hopefully, but would love to make further improvements upon requests. 

Many thanks.